# Synergistic Effect of a Defect-Free Graphene Nanostructure as an Anode Material for Lithium Ion Batteries

**DOI:** 10.3390/nano10010009

**Published:** 2019-12-18

**Authors:** Kwang Hyun Park, Byung Gon Kim, Sung Ho Song

**Affiliations:** 1Center for Artificial Low Dimensional Electronics System, Institute for Basic Science (IBS), Pohang-si, Gyeongsangbuk-do 37673, Korea; recite14@gmail.com; 2Next Generation Battery Research Center, Korea Electrotechnology Research Institute, 12 Bulmosan-ro 10beon-gil, Seongsan-gu, Changwon-si, Gyeongsangnam-do 51543, Korea; byunggonkim@keri.re.kr; 3Division of Advanced Materials Engineering, Kongju National University, Cheonan, Chungnam 32588, Korea

**Keywords:** defect free graphene, graphite intercalation compound, lithium ion battery, anode material, nanostructure

## Abstract

Graphene nanosheets have been among the most promising candidates for a high-performance anode material to replace graphite in lithium ion batteries (LIBs). Studies in this area have mainly focused on nanostructured electrodes synthesized by graphene oxide (GO) or reduced graphene oxide (rGO) and surface modifications by a chemical treatment. Herein, we propose a cost-effective and reliable route for generating a defect-free, nanoporous graphene nanostructure (*df*-GNS) through the sequential insertion of pyridine into a potassium graphite intercalation compound (K-GIC). The as-prepared *df*-GNS preserves the intrinsic property of graphene without any crystal damage, leading to micro-/nano-porosity (microporosity: ~10–50 µm, nanoporosity: ~2–20 nm) with a significantly large specific surface area. The electrochemical performance of the *df*-GNS as an anode electrode was assessed and showed a notably enhanced capacity, rate capability, and cycle stability, without fading in capacity or decaying. This is because of the optimal porosity, with perfect preservation of the graphene crystal, allowing faster ion access and a high amount of electron pathways onto the electrode. Therefore, our work will be very helpful for the development of anode and cathode electrodes with higher energy and power performance requirements.

## 1. Introduction

Since graphite, with relatively slow solid-state lithium diffusion, was initially utilized as an anode material in lithium ion batteries (LIBs), active materials with high power and energy density levels have attracted growing interest due to their potential applications in mobile electronics and the electric vehicle market. However, transition metal oxides [1,2,3,4,5], one of the most promising candidates, show relatively high redox potential values, an irreversible phase transition, and rapid capacity decay, in spite of the high theoretical capacity and low cost of these materials.

Graphene, a two-dimensional (2D) honeycomb lattice made of carbon atoms, has been enthusiastically studied, owing to its extraordinary electrical [6] and mechanical [7] properties. Furthermore, graphene has been widely used as a textual additive for the fabrication of nanostructure electrodes and as conducting agents for the improvement of the electrical conductivity between active materials. Graphene oxide (GO) and reduced graphene oxide (rGO), as well as rGO/metal oxide composites, have been widely recognized, and their electrochemical performance capabilities have been assessed as anode and cathode electrodes in lithium ion batteries [8,9,10,11,12]. For example, Jiang et al. reported a three-dimensional (3D) porous MoS_2_/GO nanostructure synthesized by a hydrothermal method, showing a significant enhancement of the rate performance [13]. Yu et al. demonstrated that a mesoporous rGO structure with a well-defined porosity, fabricated from colloidal nanocrystals (NCs), provides a considerable performance improvement, with good reproducibility in LIBs [14]. Furthermore, Lee et al. suggested a nanohole-structured architecture synthesized from gelatin-functionalized GO *via* microwave irradiation, resulting in a high rate performance with stable capacity retention [15]. Furthermore, the Daeeke, Kalantar-zadeh, and Yeo groups have suggested new methods for fabricating layered solid carbonaceous materials collected from CO_2_ conversion and their application to capacitors [16], as well as peeling off bulk 3D piezoelectric crystals (WS_2_ and MoS_2_) to 2D nanosheets by using high frequency acoustic waves [17]. However, oxygen functional groups ascribed to strong oxidation chemicals (H_2_O_2_, HNO_3_, and KMnO_4_) and imperfect crystal recovery rates have mainly led to limited performance improvements in LIBs. Recently, our group has reported the scalable fabrication of a potassium graphite intercalation compound (K-GIC) and its exfoliation to defect-free graphene [18]. However, the formation of a nanostructure with an optimal porosity level from perfect graphene remains to be achieved, required in order to realize higher energy density and power density levels with good electrochemical performance stability.

Here, we propose a simple and cost-effective method for generating a defect-free graphene nanostructure with optimal micro-/nano-porosity through the controllable expansion of K-GIC. From Raman spectroscopy and high resolution-transmission electron microscopy (HR-TEM), X-ray diffraction (XRD), and X-ray photoelectric spectroscopy (XPS) results, the synthetic *df*-GNS preserves the unique properties of pristine graphene without the introduction of additional functional groups and without structure damages on the basal plane. Our method is also scalable for the fabrication of the *df*-GNS (>1 g/batch). Regarding the electrical energy storage performance of LIBs, the *df*-GNS shows stable capacity retention at different current densities (470 mAh g^–1^ at 0.03 A g^–1^ and 350 mAh g^–1^ at 0.1 A g^–1^ after 100 cycles). Moreover, the rate capability of the *df*-GNS (retention capacity of ~18% at 5 A g^–1^) is significantly higher than that of rGO (~11% at 5 A g^–1^) and graphite (~0.1% at 5 A g^–1^). Overall, the utilization of *df*-GNS as an anode electrode, as suggested in this work, will provide a practical solution for LIBs, offering high energy density and power density levels.

## 2. Materials and Methods

### 2.1. Material and Fabrication of Defect-Free Graphene Nanostructure

Graphite powder and potassium metal were purchased from Bay Carbon, Inc., Bay, MI, USA. (SP-1 graphite powder) and Kojundo Chemical Inc., Tokyo, Japan (KKE02GB, keep in oil), respectively. For fabrication of potassium intercalation compound, a home-made apparatus was used. Graphite powder (1 g) and potassium metal (0.41 g), filled in Pyrex tubes (K metal: 10 mm in diameter, outside; Graphite powder: 6 mm in diameter, inside), were evacuated for 30 min, and the outer Pyrex tube was sealed and kept at 110 °C for 1 h in a glove box under an Ar atmosphere. Then, 200 mg of a potassium graphite intercalation compound (KC_8_) was put into 10 mL of the selected solvents and kept at 5 °C for 24 h for preparation of the *df*-GNS with optimal micro-/nano-porosity. The synthetic *df*-GNS was washed with distilled water, heated to a mild temperature (~40 °C), put through a filtration process, because of the necessity of the removal of residue potassium ions, and then dried at 80 °C for 24 h. To find the optimal conditions, experiments using various solvents and operation temperatures were conducted, and the best condition was to be kept at 0 °C when forming the *df*-GNS with a larger surface area and high pore volume porous architecture for forming porous *df*-GNS. For the solvents, hexane, toluene, tetrahydrofuran (THF), ethanol (EtOH), pyridine, dimethylformanmide (DMF), and dimethyl sulfoxide (DMSO) were chosen, according to different polarities, and were purchased from Sigma–Aldrich Chemicals Co, St. Louis, Mo, USA.

### 2.2. Electrochemical Measurement

The working electrode was fabricated by mixing the *df*-GNS, super-P, and polyvinyldifluoride (PVDF) at a weight ratio of 80:10:10 and then casting on copper foil. The electrodes were dried at 60 °C for 12 h in a vacuum atmosphere. The active material weight of the final electrodes was ~1.0 mg cm^–2^. Lithium foil (450 μm), separator (polyethylene, 2400), and electrolyte consisting solutions of LiPF_6_ in ethylene carbonate (EC)/dimethyl carbonate (DMC) (1/1 in volume) were chosen. Here, a 2032 type coin cell was selected, and the cell assembly was conducted in a glove box under an Ar atmosphere. The cycling and rate performance was tested at various current densities after a 0.03 A g^–1^ pre-cycle at a voltage range of 0.01–3.0 V.

## 3. Results and Discussion

Graphite powder was chosen to synthesize a highly ordered potassium graphite intercalation compound (K-GIC), with the overall procedure illustrated in Figure 1a. Graphite powder (1 g, 0.083 M) and potassium metal (0.407 g, 0.0104 M) was added to the Pyrex tubes and evacuated for 30 min, then sealed and kept at 110 °C for 1 h in a glove box. The potassium metals were introduced into the graphite crystals and the interlayer spacing between graphene layers was expanded from ~0.34 nm to ~0.57 nm. The potassium graphite intercalation compounds (KC_8_, gold color) were then very slowly expanded to the *df*-GNS with micro-/nano-porosity at 0 °C for 24 h in a pyridine solution. In our work, the selected method is the best preparation condition among various expansion conditions, determined after tests with different types of solvents and reaction times in the solvents. Further details of the experimental method are described in our previous work [16] and in the experimental section. Figure 1b shows digital images of the KC_8_ (before) and the *df*-GNS (after) in the pyridine solution before and after the expansion process. After 24 h at 0 °C, the KC_8_ powder in the pyridine solution changed to a slurry-like powder, with the disappearance of the gold-colored K-GIC crystals, where it was fully dispersed in the pyridine solvent. From the SEM images (Figure 1c), the morphology of the synthetic *df*-GNS showed the formation of a nanostructure with a micro-/nano-size distance towards the out-of-plane direction, without a notable size decrease from the initial graphite powder (lateral average size of ~100 µm), presenting microporosity of ~10–50 µm and nanoporosity of ~2–20 nm. In our work, pyridine, which is geometrically similar to the hexagonal lattice of graphene, was found to be the most effective expansion solvent because of its peculiar feature, where it can lead to the formation of micro-/nano-porosity due to the gradual extinction of the van der Waals (vdW) interaction between the graphene layers, with low mechanical stress via the sequential insertion of pyridine molecules into the K-GIC crystal. This result strongly supports the selection of this solvent for the expansion of the graphene layers.

Figure 2a shows typical low- and high-magnification TEM images and the selected area diffraction pattern (SAED), clearly indicating the various types of sheet-like multilayered structures and the crystal quality. The *df*-GNS sample shows a highly-ordered hexagonal lattice (inset of Figure 2a) [19]. These results prove the perfect preservation of the graphene crystal during the expansion process to the *df*-GNS without the introduction of used chemicals. From studies of the Raman spectroscopy for the graphite, KC_8_, and *df*-GNS, we observed a peak shift and broadening for the potassium graphite intercalation compound (KC_8_) and recovery of the main peaks without an increase of the D band (*df*-GNS), as shown in Figure 2b. Interestingly, the G band of the KC_8_ was broadened and exhibits two peaks at ~1520 cm^–1^ and ~1580 cm^–1^, which were caused by the coupling and interference between the phonon-electron interfaces [20], while its 2D band, known as the effective method used to determine the thickness of the graphene layer, has obviously disappeared. On the other hand, the G band of the *df*-GNS was slightly red-shifted to ~1582 cm^–1^, which could be ascribed to change of resonant vibration frequencies by the formation of the porosity, and also from its D band. Here, we confirm that none of the steps have led to any structural damages of the basal planes. Nitrogen adsorption measurement is very useful in determining the textual properties of solids, for instance, for examining the porosity and surface area. Figure 2c shows the N_2_ adsorption/desorption isotherm data measured at 77 K. The specific surface area was calculated by the Brunauer–Emmett–Teller (BET) equation. The isotherm of the *df*-GNS presents a H3 hysteresis loop according to IUPAC classification, and the specific surface area of the *df*-GNS is 281.6 m^2^ g^–1^ while that of the graphite is ~10 m^2^ g^–1^. This result suggests that porosity of the *df*-GNS is clearly developed during expansion process.

To assess *df*-GNS as an anode material in a lithium ion battery, a working electrode was fabricated by the slurry casting of a mixture of the active materials (Graphite, rGO, and *df*-GNS), super-P, and polyvinylidene fluoride (PVDF, Aldrich) at a weight ratio of 80:10:10 on a copper foil. The typical active mass loading was in the range of 1.0–1.5 mg cm^−2^. A piece of 450 μm lithium foil (Honjo metal), a polyethylene separator (Asahi Kasei), and an electrolyte containing 1 M LiPF_6_ in ethylene carbonate (EC) and dimethyl carbonate (DMC) at a 1:1 volume ratio (Panax ETEC) were chosen. Further details about the electrode preparation and electrochemical measurement methods are described in the experimental section. Figure 3 shows the typical SEM images, Raman spectrum, and survey results from XPS analyses of the graphite, the rGO, and *df*-GNS samples. For *df*-GNS, we observed the clear development of in-plane micro-/nano-porosity without a decrease in the particle-size of the initial graphite, which is in stark contrast to the outcome for the rGO sample (particle size of ~10µm with the expansion of interlayers at the edge), as shown in Figure 3a. The D band of the rGO at 1350 cm^–1^ shows a clear difference compared to that of the *df*-GNS, and the crystal structure of the rGO was notably changed by the sp^3^ hybridization of carbons, leading to extreme crystal damage (Figure 3b). X-ray photoemission spectroscopy (XPS) was used to observe the contents of oxygen or other elements. The atomic compositions of the *df*-GNS exhibit carbon of 95.7 At% and oxygen of 4.3 At% while those of the rGO show carbon 93.3 At% and oxygen 6.7 At%. while that of graphite was 98.3 At% and 1.7 At% for carbon and oxygen, as shown in Figure 3c. Additionally, the changes in the X-ray diffraction (XRD) patterns and C1s peaks from the graphite to the *df*-GNS clarify the structural evolution and changes of binding energy originating from oxygen or other elements during the procedures (Appendix A).

Figure 4a shows a schematic diagram of the synergistic effect of the anode electrode fabricated by the defect-free, porous graphene nanostructure, indicating that this process offers ion pathways and electron transfer capabilities. Figure 4b shows the galvanostatic charge-discharge profiles of the graphite and the rGO samples for the first cycle, as well as those of the *df*-GNS for the first cycle and ten cycles at a current density of 0.03 A g^–1^. The charge—discharge voltage profiles during the first cycle for the three different electrodes and ten cycles for the *df*-GNS electrode are a typical example. The graphite exhibits a clear lithiation-related plateau at ~0.01–0.25 V and the rGO shows irreversible side reactions in the voltage range of ~0.5–1.0 V. In contrast, for the *df*-GNS, a slight lithiation plateau in the range of ~0.01–0.5 V can be observed, with only a trivial irreversible side reaction. The capacities of the graphite, rGO, and *df*-GNS were 368 mAh g^–1^, 445 mAh g^–1^, and 470 mAh g^–1^, respectively. After first cycle, the reversible capacity of the *df*-GNS increased steadily with the increase of reversible capacity, and the results are well demonstrated by the synergistic effect of the larger activation sites during the charge/discharge process. The reversible capacity of the *df*-GNS was stabilized at ~470 mAh g^–1^ during the ten cycles without a noticeable capacity fade. Along with this high capacity, the columbic efficiency was also suitably stabilized from 72.3% at the first cycle to 99.0% in the ten cycles. On the other hand, the rGO sample exhibits an extremely low columbic efficiency of 46.5% in the first cycle. The low columbic efficiency in the first cycle may result from the irreversible lithium loss due to the irreversible reaction and the formation of solid electrolyte interphase (SEI) [21]. To verify the electrochemical performance, the cycling and rate performance outcomes for the graphite, rGO, and *df*-GNS were evaluated. The cycling performance of *df*-GNS at different current densities of 0.03 and 0.1 A g^–1^ (Figure 4c) was assessed, and the results were compared with those of rGO. The reversible capacities for the *df*-GNS were ~470 mAh g^–1^ at 0.03 A g^–1^ and ~350 mAh g^–1^ at 0.1 A g^–1^ after 100 cycles. Interestingly, the reversible capacity at a current density of 0.03 A g^–1^ gradually increased during the initial 40 cycles for the *df*-GNS sample. This peculiar behavior appears to be effective for facilitating lithium ion access, with a significant decrease in the resistance and an increase in the reversible capacity upon cycling. The *df*-GNS sample exhibits the best rate capability, in spite of the increasing current density (0.03 to 5 A g^–1^) when compared to that of the graphite and the rGO, as shown in Figure 4d. Specifically, the capacity of the *df*-GNS at a current density of 5 A g^–1^ is retained, with the capacity retention being ~18% compared to that of the current density at 0.03 A g^−1^ while the rGO and the graphite are 11% and 0.1%, respectively. With a return of the current rate to the initial 0.1 mA g^–1^, after 30 cycles, the *df*-GNS steadily recovers to a higher capacity rate. Our approach can pave the way for further improving and stabilizing the performance of energy storage devices such as batteries, fuel cells, and capacitors, and the current defect-free materials can be immediately applicable to active materials in other electrochemical energy storage systems suffering from similar problems.

## 4. Conclusions

We have introduced a simple and scalable route for creating micro-/nano-structured *df*-GNS, without crystal damages and the introduction of oxygen functional groups. Here, *df*-GNS was assessed as an anode electrode during electrical energy storage. From structural/chemical analyses, we have confirmed that the *df*-GNS, in an optimization condition, preserves the unique properties of graphene and forms an optimal nanostructure with a significantly larger surface area. The electrical energy storage performance of the *df*-GNS exhibits an excellent capacity retention of ~470 mAh g^–1^ at 0.03 A g^–1^ and ~350 mAh g^–1^ at 0.1 A g^–1^ after 100 cycles. Additionally, the rate capability of the *df*-GNS (~18% of initial capacity at 5 A g^–1^) is significantly enhanced compared to that of the graphite (~0.1%) and rGO (~11%). The overall results suggest that our approach will bring about the enhanced performance of an anode electrode with higher energy and power performance capabilities in electrochemical energy storage devices.

## Figures and Tables

**Figure 1 nanomaterials-10-00009-f001:**
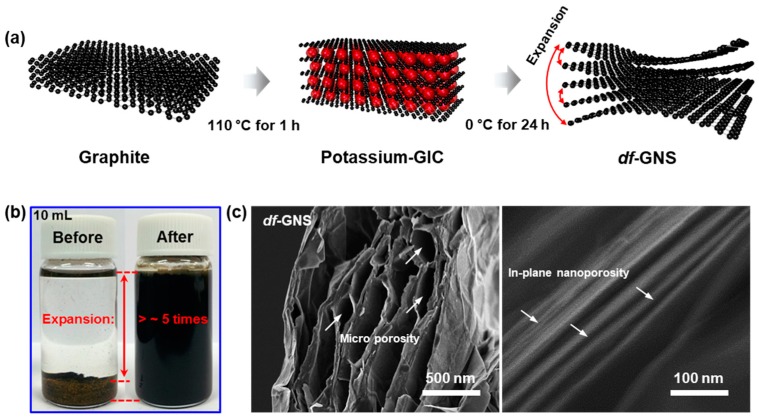
(**a**) Schematic drawing of the steps used to the defect-free graphene nanostructure. (**b**) Digital images of the potassium graphite intercalation compound (KC_8_) and the defect-free, nanoporous graphene nanostructure (*df*-GNS) dispersed in a pyridine solution. (**c**) SEM images of the *df*-GNS (microporosity shown on the left and nanoporosity shown on the right).

**Figure 2 nanomaterials-10-00009-f002:**
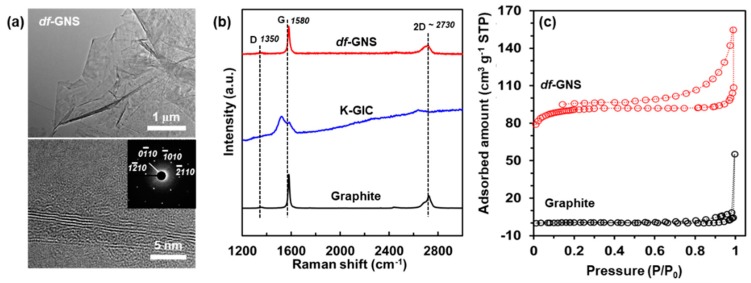
Characterizations of the graphite, KC_8_, and *df*-GNS. (**a**) Low- (top) and high- (bottom) magnification TEM images of the *df*-GNS (inset; selected area diffraction pattern (SAED) for 1–210, 0–110, –1010, and –2110). (**b**) Raman spectroscopy for the graphite, KC_8_, and *df*-GNS. (**c**) N_2_ adsorption/desorption isotherm of graphite and *df*-GNS, determined at 77 K.

**Figure 3 nanomaterials-10-00009-f003:**
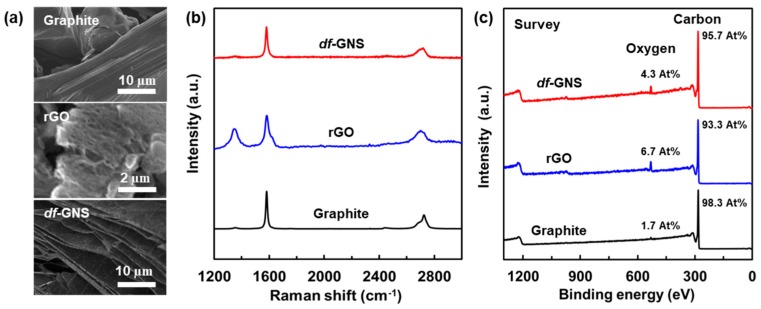
Characterizations of the graphite, rGO and *df*-GNS materials. (**a**) SEM images of graphite, rGO and *df*-GNS. (**b**) Raman spectroscopy of graphite, rGO and *df*GNS. (**c**) Chemical compositions for carbon and oxygen in the graphite, rGO, and *df*-GNS.

**Figure 4 nanomaterials-10-00009-f004:**
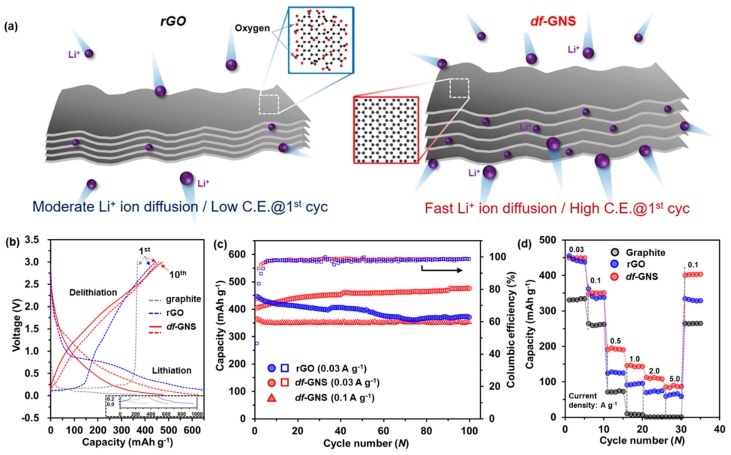
(**a**) Schematic diagram of the rGO and *df*-GNS electrodes. (**b**) Charge-discharge voltage profiles for the graphite, rGO, and *df*-GNS at a current density of 0.03 mA g^–1^. (**c**) Comparison of the cycling performance outcomes of the rGO and *df*-GNS at current densities of 0.03 A g^–1^ and 0.1 mA g^–1^ (**d**) Rate capability of the graphite, rGO, and *df*-GNS samples.

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
