# Peer review of "Synergistic Effect of a Defect-Free Graphene Nanostructure as an Anode Material for Lithium Ion Batteries"

_nanomaterials, 2019, doi:10.3390/nano10010009_

Round 1

Reviewer 1 Report

The authors synthesize a graphene-based electrode for a more efficient anode in a lithium ion battery. While the data is sound and well-presented, the novelty is low. The methods have been widely reported in the past. The actual structure is more of a supercapacitor than a battery electrode, and, thus, the title and scope are misleading. Most importantly, the authors never explain how the resulting structure is defect-free, and why that is beneficial.

Specific comments include:

The authors state that their key benefit is defect-free graphene. However, they never explain why defects are harmful. Certain defects (such as holes in the graphene sheet) may open rapid pathways for lithation and reduce diffusion costs. Other defects may increase energy of lithiation and delithiation (quantum capacitance or other similar effects). The authors do not provide either references or arguments. They also do not provide electrochemical data on a comparable defective graphene material to show how the absence of defects influences performance. The authors synthesize separated graphene sheets with significant spaces between and a structured porosity. This resembles ultracapacitor electrodes rather than a battery. Battery requires a closed crystalline structure that forces lithium ions to intercalate into the structure. Porosity relies on electrostatic adsorption of ions under the influence of an electric field. The latter process is much quicker, and is likely the mechanism here. The absence of staging (distinct sharp inflection points) shown in Figure 4b is a clear giveaway. This is not a battery but a capacitor. What evidence do the authors have that there are no defects in the graphene sheet? They do not provide crystallize size. All shown data is typical for multilayered graphene that has been widely reported over the last decade, so the benefit or uniqueness of their approach are unclear.

Author Response

Response: we are grateful to the reviewer for evaluating our manuscript. We also doubly thank you for your comments of our manuscript and appreciate your concerns about novelty compared with previous studies including anode electrodes based on graphene oxides and reduced graphene oxides in the lithium ion battery. Our work is fundamentally different from the earlier works and a significant advancement in the whole field of the battery anodes because the defect-free graphene nanosheet (df-GNS) forming micro/nano porosity (SEM image and N2 adsorption/desorption isotherm, as shown in Figure 1c and Figure 2c) without any crystal damages on the basal plane of graphene (clear diffraction pattern and absence of the D band, as shown in Figure 2a and Figure 2b) is synthesized, and the as-prepared df-GNS is the first demonstration as anode electrode in lithium ion battery. The lateral crystal size of the df-GNS is quite similar with that of initial graphite powder (lateral avg. size: ~100 µm, SP-1 graphite powder) without significant decrease of the size, and the statement is added on the page 3, line 116 in the revised manuscript. Moreover, our synthetic method shows complete preservation of initial graphene properties, even after exfoliation to the isolated graphene flake with mono/-few layer (Park et. al., Nano Lett., 2014, 14, 4306). This result indicates that there are no crystal damages in our expansion procedures.

Furthermore, reviewer comments synergistic effect of battery performance ascribed to creating holes or protrusions in graphene, which can lead to rapid pathways for Lithiation/Delithiation with reducing diffusion costs as well as enhanced Li storage site (quantum capacitance). We totally agree with the reviewer that the hole or protrusion creation of the graphene basal plane can play a significant role for performance enhancement in lithium ion battery (Wang et al., Chemsuschem, 2013, 6, 56; Schiros et al., Nano Lett., 2012, 12, 4025; Pei et al., Energy Environ. Sci., 2017, 10, 742; Yang et. al., Energy Environ. Sci., 2017, 10, 979). In this regard, we would like to stress fabrication of the df-GNS nanostructure with micro-/nano-size distance toward out-of-plane direction, which can allow faster ion access and high electron pathways. Besides, the df-GNS based electrode shows notable enhancement of capacity and rate capability as well as cycle stability.  

We also thank the reviewer for invaluable comments about supercapacitors. It is certainly true that graphene based structures with large porosity have been evaluated for the performance in supercapacitors. However, we would like to mention about all the previous works. Actually, various types of the nanostructures prepared from the graphene oxides and reduced graphene oxides have been widely studied as anodes of lithium rechargeable batteries as well as electrode of capacitors (Xie et al., J. Power. Sources, 2015, 273, 754; Kim et al., Sci. Rep., 2014, 4, 5278; Zhang et al., Carbon, 2014, 74, 153; Zhang et al., J. Power Sources, 2013, 241, 619; Reddy et al. ACS Nano, 2010, 4, 6337; Yoo et al., Nano Lett., 2008, 8, 2277; Wang et al., Chem. Mater, 2009, 21, 2604). For this reason, the performance test of the df-GNS ascribed to modulation of the pore size and distribution as electrode of non-pseudo and pseudo capacitors is underway. We believe that the df-GNS is a promising candidate for realizing high performance lithium ion batteries as well as supercapacitors.

Reviewer 2 Report

The authors showed defect-free, nanoporous graphene by sequential insertion of pyridine into potassium graphite. They demonstrated that the defect free material is graphene-like with high cystallinity with micro-/nano-porosity. They studied the electrochemical performance of the materials as the anode electrode with high capacity, rate capability, and and cycle stability. 

The work is strong and certainly merit publication 

I only have the following minor suggestions 

1- Discuss the new methods for preparing graphene and 2D sheets including Nature Communications 10 (1), 865, 2019 and Advanced Materials 30 (20), 1704756, 2018

2- In the abstract instead of improved used enhanced

3- Expand the conclusion with some future suggestions 

Author Response

Discuss the new methods for preparing graphene and 2D sheets including Nature Communications 10 (1), 865, 2019 and Advanced Materials 30 (20), 1704756, 2018

Response: We thank the reviewer for carefully reviewing our manuscript and delivering helpful comments. According to the reviewer’s suggestion, we discuss the new method for fabricating layered solid carbonaceous materials collected from CO2 conversion and its application to capacitor (Esrafilzadeh et al., Nat. Commun., 2019, 10, 865) as well as peeling off bulk 3D piezoelectric crystals (WS2 and MoS2) to 2D nanosheets by using high frequency acoustic waves (Ahmed et al., Adv. Mater., 2018, 30, 1704756). The statement for new methods is added on the page 2, line 51 in introduction of the revised manuscript (ref 16 and ref 17).

Add sentence

Furthermore, Daeeke, Kalantar-zadeh, and Yeo groups suggested new methods for fabricating layered solid carbonaceous materials collected from CO2 conversion and its application to capacitor [16] as well as peeling off bulk 3D piezoelectric crystals (WS2 and MoS2) to 2D nanosheets by using high frequency acoustic waves [17].

In the abstract instead of improved used enhanced

Response: Thank you for your kind suggestion. The "improved" word is revised to "enhanced" on the page 1, line 23 of the revised manuscript.

Expand the conclusion with some future suggestions

Response: Thank you for your kind suggestion. According to your comment, we add the statement for some future suggestion on the page 5 line 198 of the revised manuscript.

Add sentence

Our approach can pave the way for further improving and stabilizing the performance of energy storage devices such as batteries, fuel cells, and capacitors, and the current defect-free materials can be immediately applicable to active materials in other electrochemical energy storage systems suffering from similar problems.

Round 2

Reviewer 1 Report

Authors improved the manuscript.